# Logical Specifications-guided Dynamic Task Sampling
# for Reinforcement Learning Agents

**Primary Keywords:** *(2) Learning; (8) Knowledge Representation/Engineering*

## Abstract

Reinforcement Learning (RL) has made significant strides in enabling artificial agents to learn diverse behaviors. However, learning an effective policy often requires a large number of environment interactions. To mitigate sample complexity issues, recent approaches have used high-level task specifications, such as Linear Temporal Logic ($LTL_f$) formulas or Reward Machines (RM), to guide the learning progress of the agent. In this work, we propose a novel approach, called Logical Specifications-guided Dynamic Task Sampling (LSTS), that learns a set of RL policies to guide an agent from an initial state to a goal state based on a high-level task specification, while minimizing the number of environmental interactions. Unlike previous work, LSTS does not assume information about the environment dynamics or the Reward Machine, and dynamically samples promising tasks that lead to successful goal policies. We evaluate LSTS on a gridworld and show that it achieves improved time-to-threshold performance on complex sequential decision-making problems compared to state-of-the-art RM and Automaton-guided RL baselines, such as Q-Learning for Reward Machines and Compositional RL from logical Specifications (DIRL). Moreover, we demonstrate that our method outperforms RM and Automaton-guided RL baselines in terms of sample-efficiency, both in a partially observable robotic task and in a continuous control robotic manipulation task.

## 1 Introduction

Agents are now capable of learning optimal control behavior for a broad spectrum of tasks, ranging from Atari games (Gao and Wu 2021) to robotic manipulation tasks (Nguyen and La 2019), thanks to recent advancements in Reinforcement Learning (RL). Despite the progress made in RL, learning an optimal task policy using model-free RL techniques still suffers from sample complexity issues because of sparse reward settings and unknown transition dynamics (Lattimore, Hutter, and Sunehag 2013). These challenges further intensify in long-horizon settings, where the agent needs to perform a series of correct sequential decisions to achieve the goal. Additionally, certain tasks (such as - robot needs to make dinner only if it bought groceries in the afternoon) require the agent to *encode* and *remember* its episodic history (whether the groceries were bought) in order to solve the task effectively. To alleviate this issue in complicated tasks, several lines of work have explored representing the goal using an intricately shaped reward function that guides the agent toward the goal (Grzes 2017). However, generating such a reward function requires the human engineer to assign 'importance' weights to various aspects of the task, which is time consuming and assumes knowledge on which aspects of the task are important. Poorly engineered reward functions can lead to local optima, where the agent learns to satisfy only a subset of goals and ignores the rest.

Recent research has investigated representing the goal using high-level specification languages, such as finite-trace Linear Temporal Logic ($LTL_f$) (De Giacomo and Vardi 2013), Reward Machines (RM) (Icarte et al. 2022), SPECTRL (Jothimurugan, Alur, and Bastani 2019) that allow defining the goal of the task using a graphical representation of sub-tasks. The high-level objective is known before commencing the task, and the graphical representation allows the agent to achieve easier sub-goals initially, and build upon them to achieve complex goals. Encoding the task using a graphical structure allows us to tackle the problem in a Markovian manner by tracking the history as a part of the state space (Afzal et al. 2023), thereby allowing the agent to keep track of its episodic history. For instance, if the task for a robot is to reach kitchen and then make dinner, the graphical structure of the task obtained from the high-level specification allows the agent to reason whether it has reached the kitchen before it can commence its policy for making dinner. RM approaches still require human guidance in defining the reward structure of the machine, which is dependent on knowing how much reward should be assigned for accomplishing each sub-goal. The process of designing the reward structure assumes that the human engineer is aware of how much should reward should the agent receive when it accomplishes the sub-goals in particular order. This assumption is infeasible in scenarios when the structure of the environment or the exact order in which the sub-goals must be achieved is unknown in advance. In contrast, our method does not require access to the reward structure.

Another method, Compositional RL from Logical Specifications (DIRL) (Jothimurugan et al. 2021) mitigates the reward assignment issue by using Dijkstra's algorithm to determine which sub-tasks (edges) should be explored in the SPECTRL DAG graph (Jothimurugan, Alur, and Bastani 2019) in order to learn policies to reach nodes in the DAG

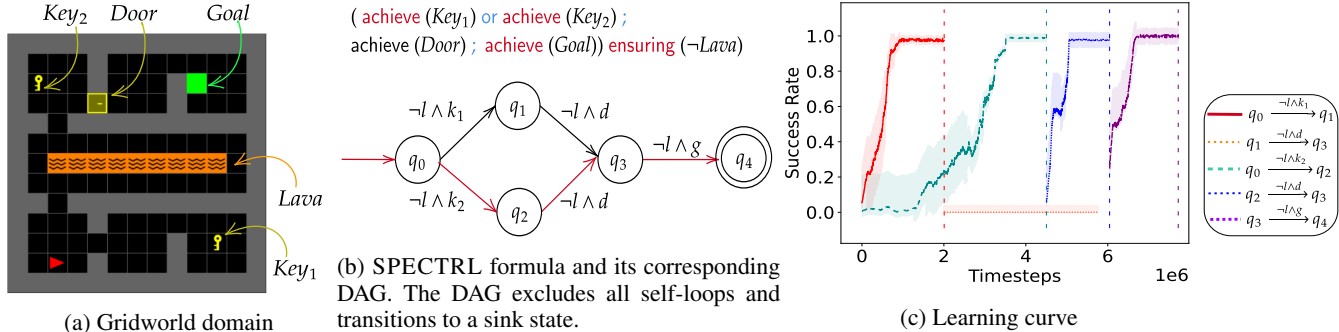

(a) Gridworld domain

(b) SPECTRL formula and its corresponding DAG. The DAG excludes all self-loops and transitions to a sink state.

(c) Learning curve

Figure 1: (a) Gridworld domain and descriptors. The agent (red triangle) needs to collect one of the keys and open the door to reach the goal; (b) The SPECTRL formula for the task and its DAG. Formulas $l$, $k_1$, $k_2$, $d$ and $g$ correspond to $Lava$, $Key_1$, $Key_2$, $Door$ and $Goal$ respectively; (c) Learning curves for individual sub-tasks (averaged over 10 trials) generated using *LSTS*. The path chosen by *LSTS* is highlighted in red in Fig.1(b)

that yield the highest success rate. DıRL requires the agent to learn RL policies for satisfying *all* outgoing edge propositions (each edge encodes a sub-task) from such nodes. However, this approach requires the agent to explore a sub-task for a manually specified number of interactions, which requires knowledge about the task complexity. DıRL ends up spending a lot of interactions learning unproductive policies as some sub-tasks can be unpromising, yet the agent has to spend the defined number of interactions learning a policy for the sub-task. Unlike DıRL, our approach is sample-efficient as it finds unpromising sub-tasks based on the learning progress of the sub-tasks, and discards them; saving costly interactions and converging to a successful policy faster. This problem of minimizing the overall number of interactions while learning a set of successful policies is non-trivial as the problem equates to finding the shortest path in a graph whose true edge weights are unknown *a priori* (Szepesvári 2004). In our case, the edge weight denotes the total number of environmental interactions required by the agent to learn a successful policy for the sub-task encoded by the edge, in which the agent must induce a visit to a state where certain properties hold true. And, we can sample interactions for a sub-task only if we have a policy to reach the edge's source node from the start node of the graph, making the learning process more sample-inefficient.

To address the above challenges, we present Logical Specifications-guided Dynamic Task Sampling (*LSTS*). We begin with a high-level objective represented using SPECTRL specification formulas which can equivalently be represented using directed acyclic graphs (DAG) (Jothimurugan, Alur, and Bastani 2019). The DAG structure encodes memory, helping the agent understand what events of interest have occurred in the past, and which events must occur to reach the accepting states. Our key insight is to learn RL policies for sub-tasks defined using the edges of the DAG. Specifically, the agent transitions from the node $q$ to $p$ in the DAG when the propositional logic formula labeling the edge $(q, p)$ evaluates to true. We use the set of propositional logic formulas labeling the outgoing edges from a given node in DAG to define sub-tasks. The trajectory induced by a suc-

cessful RL policy for the sub-task enables the agent's high-level state in the DAG to transition from the source node to the destination node of the edge defining the sub-task. We employ an adaptive Teacher-Student learning strategy, where, (1) the Teacher agent uses its high-level policy along with exploration techniques to actively sample a sub-task for the Student agent to learn. The high-level policy considers the DAG representation and the Student agent's expected performance on all the sub-tasks, aiming to satisfy the high-level objective in the fewest number of interactions, and (2) the Student agent interacts with the environment for a few steps (much fewer than the interactions required to learn a successful policy for the sub-task) while updating its low-level RL policy for the sampled sub-task. The Teacher observes the Student's performance on these interactions and updates its high-level policy. Steps (1) and (2) continue alternately until the Student agent learns a set of successful policies that guide the agent to reach a goal state.

**Running example:** As an example, let us look at the environment shown in Fig. 1a. The goal for the agent is to collect *any* of the two *Keys*, followed by opening the *Door* and then reaching the *Goal* while avoiding the *Lava* at all times. The task's high-level objective ($\phi$) is represented using the SPECTRL formula and its corresponding DAG representation $\mathcal{G}_\phi$ in Fig. 1b. The DAG does not contain information about the environment configuration, such as: the optimal number of interactions required to reach $Door$ from $Key_1$ are much higher compared to the interactions required to reach $Door$ from $Key_2$, making the $Key_1$ to $Door$ trajectory sub-optimal. Hence, it is crucial to prevent any additional interactions the agent spends in learning a policy for the sub-task defined by the edge $q_1 \xrightarrow{\neg l \wedge d} q_3$ as the path $q_0 \xrightarrow{\neg l \wedge k_1} q_1 \xrightarrow{\neg l \wedge d} q_3 \xrightarrow{\neg l \wedge g} q_4$ will always be sub-optimal. In our proposed approach *LSTS*, the Student agent begins with the *aim* of learning two distinct RL policies: $\pi_1$ for the task of visiting $Key_1$ and $\pi_2$ for the task of visiting $Key_2$, both avoiding $Lava$. The Teacher agent initially samples evenly from these two sub-tasks for the Student but later biases its sampling toward the sub-task on which the

Student agent shows higher learning potential. Once the Student agent learns a successful policy for one of the sub-tasks (let's say the learned policy $\pi_1^*$ corresponding to the sub-task defined by the transition $q_0 \xrightarrow{\neg l \wedge k_1} q_1$), the Teacher does not sample that task anymore, identifies the next task(s) for the Student using the DAG representation, and appends them to the set of tasks it is currently sampling (in this case, the only next task is: $q_1 \xrightarrow{\neg l \wedge d} q_3$). Since the Student agent only has access to the state distribution over $q_0$, it follows the trajectory given by $\pi_1^*$ to reach a state that lies in the set of states where the proposition $\neg Lava \wedge Key_1$ holds true before commencing its learning for the policy ($\pi_3$) for $q_1 \xrightarrow{\neg l \wedge d} q_3$. If the Student agent learns the policies $\pi_2^*$ for satisfying the sub-task defined by $q_0 \xrightarrow{\neg l \wedge k_2} q_2$ and $\pi_4^*$ for $q_2 \xrightarrow{\neg l \wedge d} q_3$ before learning $\pi_3$, it effectively has a set of policies to reach the node $q_3$. Thus, the Teacher will now only sample the next task for the Student in the DAG representation $q_3 \xrightarrow{\neg l \wedge g} q_4$, as learning RL policies for paths that reach $q_3$ are effectively redundant. This process continues iteratively until the Student agent learns a set of policies that reach the goal node ($q_4$) from the start node ($q_0$). The learning curves in Fig. 1c empirically validate the running example. As evident from the learning curves, the Student agent learns policies for the path $q_0 \xrightarrow{\neg l \wedge k_2} q_2 \xrightarrow{\neg l \wedge d} q_3 \xrightarrow{\neg l \wedge g} q_4$ that produce trajectories to reach the goal node $q_4$ from the initial node $q_0$, without excessively wasting interactions on the unpromising sub-task $q_1 \xrightarrow{\neg l \wedge d} q_3$. The dashed lines in Fig. 1c signify the interactions at which a task policy converged.

The dynamic task sampling strategy promotes *LSTS* to achieve sample-efficient learning on complex tasks by identifying unpromising tasks and discarding them, saving costly interactions. Our empirical results show that *LSTS* reduces environmental interactions by orders of magnitude compared to state-of-the-art Specifications-Guided RL Baseline DIRL, Reward Machine-based baselines QRM (Icarte et al. 2018), GSRS (Camacho et al. 2018), and curriculum learning baseline TSCL (Matiisen et al. 2020). We also evaluate *LSTS^{ct}*, a modified algorithm that further improves sample efficiency by continuing exploration on a new sub-task once a goal state for a sub-task is reached. We perform evaluation on two robotic navigation and manipulation tasks and demonstrate that *LSTS* reduces the number of interactions by orders-of-magnitude when compared to state-of-the-art automaton-guided RL baselines.

## 2 Related Work

**Automaton-guided RL approaches** utilize temporal logic-based language specifications to define tasks (Toro Icarte et al. 2018; Bozkurt et al. 2020; Xu and Topcu 2019; Alur et al. 2022). Separating policies for task sub-goals aids in abstracting knowledge that can be utilized in novel tasks (Icarte et al. 2018), without reliance on a dense reward function. Another technique is to shape the reward in proportion to the distance from the accepting node in the automaton (Camacho et al. 2018); however, this often leads to suboptimal reward settings. Augmenting the reward function with Monte

Carlo Tree Search helps mitigate this issue (Velasquez et al. 2021). This approach requires the ability to plan-ahead in the environment, which is not always feasible. Automaton-guided RL has been used to aid navigational exploration for robotic domains (Cai et al. 2023) and for multi-agent settings (Hammond et al. 2021). Generating a curriculum given the high-level objective (Shukla et al. 2023) requires access to the Object-Oriented MDP (Diuk, Cohen, and Littman 2008), which cannot be obtained if environment details are not known in advance. DIRL interleaves high-level planning with RL to learn a policy for each edge, which overcomes challenges arising from poor representations (Jothimurugan et al. 2021). This approach becomes inefficient in terms of number of interactions, as it requires the agent to act for a predetermined number of interactions, even if learning the task does not show any promise. Unlike previous works, in this paper, we propose an logical specifications-guided dynamic task sampling approach that does not require access to the environment dynamics or the Reward Machine, and efficiently samples tasks that show promise toward the high-level objective, saving interactions on unpromising tasks.

**Teacher-Student algorithms** (Matiisen et al. 2020) have been previously studied in Curriculum Learning literature (Narvekar et al. 2020; Shukla et al. 2022) and in the Intrinsic Motivation literature (Oudeyer and Kaplan 2009). The idea is to have the Teacher propose those tasks to Student on which the Student shows most promise. This strategy helps Student learn simpler tasks first, transferring its knowledge to complex tasks. The technique reduces the overall number of interactions necessary to learn a successful policy. These approaches tend to optimize a curriculum to learn a single policy, which does not scale well to temporally-extended tasks. Instead, we propose an Logical Specifications-guided Teacher-Student learning strategy that learns a policies for promising automaton transitions, promoting sample-efficient training compared to the baselines.

## 3 Theoretical Framework

**Episodic MDP.** An episodic labeled Markov Decision Process (MDP) $M$ is a tuple $(\mathcal{S}, \mathcal{A}, P, R, \mathcal{S}_0, \gamma, K, \mathcal{P}, L)$, where $\mathcal{S}$ is the set of states, $\mathcal{A}$ is the set of actions, $P(s'|s,a)$ denotes the transition probability of reaching state $s' \in \mathcal{S}$ from $s \in \mathcal{S}$ using action $a \in \mathcal{A}$, $R : \mathcal{S} \times A \times \mathcal{S} \to \mathbb{R}$ is the reward function, $\mathcal{S}_0$ is the initial state distribution, $\gamma \in [0, 1]$ is the discount factor, $K$ is the maximum number of interactions in any episode, $\mathcal{P}$ is a set of predicates, and $L : \mathcal{S} \to 2^{\mathcal{P}}$ is a labeling function that maps a state $s \in \mathcal{S}$ to a subset of predicates that are true in that state. In every interaction, the agent observes the current state $s$ and selects an action $a$ according to its policy function $\pi(a|s, \theta)$ with parameters $\theta$. The MDP transitions to a new state $s' \in \mathcal{S}$ with probability $P(s' \mid s, a)$. The agent's goal is to learn an *optimal policy* $\pi^*$ that maximizes the discounted return $G_0 = \sum_{k=0}^{K} \gamma^k R(s_k', a_k, s_k)$ until the end of the episode, which occurs after at-most $K$ interactions.

**High level specification language**: In our framework, we adopt the specification language SPECTRL to articu-

late reinforcement learning tasks (Jothimurugan, Alur, and Bastani 2019). A specification $\phi$ in SPECTRL is a logical formula applied to trajectories, determining whether a given trajectory $\zeta = (s_0, s_1, \ldots)$ successfully accomplishes a desired task. Mathematically, $\phi$ can be depicted as a function $\phi : \mathcal{Z} \to \mathbb{B}$, where $\mathbb{B} = \{\textsc{True}, \textsc{False}\}$ and $\mathcal{Z}$ is the set of all trajectories.

Formally, a specification is defined over a set of *atomic predicates* $\mathcal{P}_0$. Each $p \in \mathcal{P}_0$ is associated with a function $f_p : S \to \mathbb{B}$. The agent's MDP state $s$ satisfies $p$ (denoted by $s \models p$) when $f_p(s) = \texttt{True}$ (in other words, $p \subseteq L(s)$).

The set of *predicates* $\mathcal{P}$ comprises conjunctions and disjunctions of atomic predicates $\mathcal{P}_0$. A predicate $b \in \mathcal{P}$ follows the grammar $b ::= p \mid (b_1 \wedge b_2) \mid (b_1 \vee b_2)$, where $p \in \mathcal{P}_0$. Each predicate $b \in \mathcal{P}$ corresponds to a function $f_b : S \to \mathbb{B}$ defined naturally over Boolean logic.

The syntax of SPECTRL specifications is given by

$$\phi ::= \texttt{achieve } b \mid \phi_1 \texttt{ ensuring } b \mid \phi_1; \phi_2 \mid \phi_1 \texttt{ or } \phi_2,$$

where $b \in \mathcal{P}$. Here, $\texttt{achieve}$ and $\texttt{ensuring}$ correspond to 'eventually' and 'always' operators in temporal logic. Each specification $\phi$ corresponds to a function $f_\phi : \mathcal{Z} \to \mathbb{B}$, and $\zeta \in \mathcal{Z}$ satisfies $\phi$ (denoted $\zeta \models \phi$) if $f_\phi(\zeta) := \textsc{True}$. The SPECTRL semantics for a finite trajectory $\zeta$ of length $t$ are:

$$\zeta \models \texttt{achieve } b \quad \text{if } \exists i \le t, \ s_i \models b \ (\text{or } b \subseteq L(s)) \quad (1)$$

$$\zeta \models \phi \texttt{ ensuring } b \quad \text{if } \exists i \le t, \ s_i \models b \quad (2)$$

$$\zeta \models \phi_1; \phi_2 \quad \text{if } \exists i < t, \ \zeta_{0:i} \models \phi_1 \text{ and } \zeta_{i+1:t} \models \phi_2 \quad (3)$$

$$\zeta \models \phi_1 \texttt{ or } \phi_2 \quad \text{if } \zeta \models \phi_1 \text{ or } \zeta \models \phi_2 \quad (4)$$

Intuitively, the condition (1) signifies that the trajectory should *eventually* reach a state where the predicates $b$ hold true. The condition (2) signifies that the trajectory should satisfy specification $\phi$ while *always* remaining in states where $b$ holds true. The condition (3) signifies that the trajectory should sequentially satisfy $\phi_1$ and then $\phi_2$. The condition (4) signifies that the trajectory should satisfy either $\phi_1$ or $\phi_2$. A trajectory $\zeta$ satisfies $\phi$ if there is a $t$ such that the prefix $\zeta_{0:t}$ satisfies $\phi$.

Furthermore, each SPECTRL specification $\phi$ is *guaranteed* to have an equivalent directed acyclic graph (DAG), called an abstract graph. An *abstract graph* $\mathcal{G} = (Q, E, q_0, F, \beta, \mathcal{Z}_{safe}, \kappa)$ is a directed acyclic graph (DAG) with nodes $Q$, (directed) edges $E \subseteq Q \times Q$, initial node $q_0 \in Q$, final nodes $F \subseteq Q$, subgoal region map $\beta : Q \to 2^S$ such that for each $q \in Q$, $\beta(q)$ is a subgoal region and *safe trajectories* $\mathcal{Z}_{safe} = \bigcup_{e \in E} \mathcal{Z}_{safe}^e$ where $\mathcal{Z}_{safe}^e \subseteq \mathcal{Z}_f$ denotes the safe trajectories for edge $e \in E$. Intuitively, $(Q, E)$ is a standard DAG, and $q_0$ and $F$ define a graph reachability problem for $(Q, E)$. Furthermore, $\beta$ and $\mathcal{Z}_{safe}$ connect $(Q, E)$ back to the original MDP $M$; in particular, for an edge $e = q \to q'$, $\mathcal{Z}_{safe}^e$ is the set of trajectories in the MDP $M$ that can be used to transition from $\beta(q)$ to $\beta(q')$[1]. The function $\kappa$ labels each edge $e = q \to q'$ with the predicates $b_e$ (labeled edge denoted as $e := q \xrightarrow{b_e} q'$). The agent transitions from $q$ to $q'$ when the states $s_i, s_{i+j}$ in the agent's trajectory $\zeta$ satisfy $s_i \subseteq \beta(q)$ and $b_e \subseteq L(s_{i+j})$ and $j \ge 0$.

---

[1] See DIRL (Jothimurugan et al. 2021) for more details

Given a SPECTRL specification $\phi$, we can construct an abstract graph $\mathcal{G}_\phi$ such that, for every trajectory $\zeta \in \mathcal{Z}$, we have $\zeta \models \phi$ if and only if $\zeta \models \mathcal{G}_\phi$. Thus, we can solve the reinforcement learning problem for $\phi$ by solving the reachability problem for $\mathcal{G}_\phi$. As described below, we leverage the structure of $\mathcal{G}_\phi$ in conjunction with reinforcement learning to do so. In summary, SPECTRL specifications provide a powerful and expressive means to define and evaluate reinforcement learning tasks. It allows users to specify complex conditions and requirements for successful task completion, enabling a nuanced approach to learning from specifications.

**Problem Formulation.** Given an MDP $M$ with unknown transition dynamics and a SPECTRL formula $\phi$ representing the high-level objective of the agent, let $\mathcal{G}_\phi$ be the DAG representing the language of $\phi$. Let $\textsf{Paths}(q, X)$ be the set of all paths in the DAG originating in $q$ and terminating at a node in $X \subseteq Q$. The aim of this work is to learn a set of policies $\pi_i^*$, $i = 0, \ldots, n-1$, with the following three properties: (i) Following $\pi_0^*$ results in a trajectory in the MDP that induces a transition from $q_0$ to some state $q_1 \in Q$ in the DAG, following $\pi_1^*$ results in a trajectory in MDP that induces a transition from $q_1$ to some state $q_2 \in Q$ in the DAG, and so on. (ii) The resulting path $q_0 q_1 \ldots q_n$ in the DAG terminates at a final node, *i.e.*, $q_n \in F$, with probability greater than a given threshold, $\eta \in (0, 1)$. (iii) The total number of environmental interactions spent in exploring and learning sub-task policies are minimized.

## 4 Methodology

**Sub-task definiton:** Given the DAG $\mathcal{G}_\phi$ representing the language of $\phi$, we define a set of sub-tasks based on the edges of the DAG. Intuitively, given any MDP state $s \in \mathcal{S}$ and a DAG node $q \in Q$, a sub-task defined by an edge from node $q$ to $p \in Q$ defines a reach-avoid objective for the agent represented by the SPECTRL formula,

$$\textsf{Task}(q, p) := \texttt{achieve}(b_{(q,p)}) \texttt{ ensuring} \left( \bigwedge_{r \in \textsf{Sc}(q), r \ne p} \neg b_{(q,r)} \right)$$

where $b_{(q,p)}$ is the propositional formula labeling the edge from $q$ to $p$ in the DAG and $\textsf{Sc}(q)$ is the set of successors of node $q$ in DAG. For example, in Fig. 1b, the propositional formula labeling the edge from $q_0$ to $q_1$ is $b_{(q_0, q_1)} = \neg l \wedge k_1$. When $e = (q, p)$, we use $\textsf{Task}(e)$ instead of $\textsf{Task}(q, p)$ and $b_e$ instead of $b_{(q,p)}$ for notational convenience.

Each sub-task $\textsf{Task}(q, p)$ defines a problem to learn a policy $\pi_{(q,p)}^*$ such that, given any MDP state $s_0 \in \mathcal{S}$, following $\pi_{(q,p)}^*$ results in a trajectory $s_0 s_1 \ldots s_n$ in MDP that induces the path $qq \ldots qp$ in the DAG. That is, the agent's high-level DAG state remains at $q$ until it transitions to $p$. While constructing the set of sub-tasks, we omit transitions that lead to a 'sink' state (from which final states are unreachable).

Given the MDP $M$ with unknown transition dynamics and the SPECTRL objective, $\phi$, we first translate $\phi$ to its corresponding directed acyclic graphical (DAG) representation $\mathcal{G}_\phi = (Q, E, q_0, F, \beta, \mathcal{Z}_{safe}, \kappa)$. Next, we define the

set of sub-tasks. For this, we consider the edges that lie on some path in the DAG from $q_0$ to some node in $F$. This is because any path that does not visit $F$ leads to a sink state from which the objective cannot be satisfied. Such edges is identified using breadth-first-search (Moore 1959).

**LSTS Initialization:** The algorithm for *LSTS* is described in Algo. 1. We begin by initializing the following (lines 2-4): (1) Set of: Active Tasks AT, Learned Tasks LT, Discarded Tasks DT; (2) A Dictionary $\Pi$ that maps a sub-task Task$(e)$ corresponding to edge $e$ of DAG $\mathcal{G}_\phi$ to a RL policy $\pi_e$; (3) A Dictionary representing the Teacher Q-Values $Q$ by mapping Task$(e)$ to a numerical value associated with Task$(e)$.

Firstly, we convert $\mathcal{G}_\phi$ into an Adjacency Matrix $\mathcal{X}$ (line 6), and find the tasks corresponding to the set all the outgoing edges $\overline{E}_{q_0} \subseteq E$ from the initial node $q_0$ (line 7). Satisfying the edge's predicates $b_{(q_0,q_1)} \in \kappa(\overline{E}_{q_0})$ represent a reachability sub-task $M'$ where each goal state $s \in \mathcal{S}_f^{M'}$ of $M'$ satisfy the condition $b_{(q_0,q_1)} \subseteq L(s)$. The Student agent receives a positive reward for satisfying $b_{(q_0,q_1)}$ and a small negative reward at all other time steps. The state and action space, and the transition dynamics of $M'$ are equivalent to $M$. To complete the sub-task, the Student agent must learn a RL policy $\pi^*_{(q,p)}$ that ensures a visit from $q$ to $p$ with probability greater than a predetermined threshold ($\eta$). Moreover, the policy must also avoid triggering unintended transitions in the DAG. For instance, while picking up $Key_1$, the policy must not inadvertently pick up $Key_2$.

**Teacher-Student learning:** We set the Teacher Q-Values for all the sub-tasks corresponding to edges in AT (i.e., tasks corresponding to $\overline{E}_{q_0}$) to zero (line 8). We formalize the Teacher's goal of choosing the most promising sub-task as a *Partially Observable MDP* (Kaelbling, Littman, and Cassandra 1998), where the Teacher does not have access to the entire Student agent state but only to the Student agent's performance on a sub-task (e.g. success rate or average returns), and as an action, chooses a sub-task Task$(e) \in$ AT the Student agent should train on next. In this POMDP setting, each Teacher action has an Q-Value associated with it. Intuitively, higher Q-Values correspond to tasks on which the Student agent is more successful, and the Teacher should sample such tasks at a higher frequency to satisfy $\phi$ (reach a goal node) in fewest overall interactions.

(A) Given the Teacher Q-Values, we sample a sub-task Task$(e) \in$ AT using the $\epsilon-$greedy exploration strategy (line 10), and (B) The Student agent trains on task Task$(e)$ using the RL policy $\Pi[e]$ for '$x$' interactions (line 11). In one Teacher timestep, the Student trains for $x$ environmental interactions. Here, $x << $ total number of environmental interactions required by the Student agent to learn a successful RL policy for Task$(e)$, since the aim is to keep switching to a sub-task that shows highest promise. (C) The Teacher observes the Student agent's average return $g_t$ on these $x$ interactions, and updates its Q-Value for Task$(e)$ (line 12):

$$Q[e] \leftarrow \alpha(g_t) + (1 - \alpha)Q[e] \quad (5)$$

where $\alpha$ is the Teacher learning rate, $g_t$ is the average

Student agent return on Task$(e)$ at the Teacher timestep $t$. As the learning advances, $g_t$ increases as $t$ increases, and hence we use a constant parameter $\alpha$ to tackle the non-stationary problem of a moving return distribution. Several other algorithms could be used for the Teacher strategy (e.g., UCB or Thomspson Sampling). Steps (A), (B), (C) continue successively until the policy for *any* Task$(e) \in$ AT task converges.

**Sub-task convergence criteria:** We define Student agent's RL policy for Task$(q, p)$ to be converged (line 13) if a trajectory $\zeta$ produced by the policy triggers the transition with probability $\Pr_{\zeta \in \mathcal{Z}}[\zeta$ satisfies Task$(q,p)] \geq \eta$ and $\Delta(g_t, g_{t-1}) < \tau$ where $\eta$ is the expected performance and $\tau$ is a small numerical value. Intuitively, a converged policy attains an average success rate $\geq \eta$ and has not improved further by maintaining constant average returns. Like all other RM and automaton-based approaches, we assume access to the labeling function $L$ to examine if the trajectory $\zeta$ satisfies the formula $b_{(q,p)}$ by checking if the final environmental state $s$ of the trajectory satisfies the condition $b_{(q,p)} \subseteq L(s)$. The sub-goal regions need not be disjoint, i.e., the same state $s$ can satisfy propositions for multiple DAG nodes. Once a policy for the Task$(q, p)$ converges, we append Task$(q, p)$ to the set of Learned Tasks LT and remove it from the set of Active Tasks AT (line 14). In order to ensure that the learned sub-task does not get sampled any further, we set the Teacher Q-value for this sub-task to $-\infty$ (line 15).

**Discarding unpromising sub-tasks:** Once we have a successful policy for the Task$(q, p)$ (the transition $q \xrightarrow{b_{(q,p)}} p$), we determine the sub-tasks that can be discarded (line 16). We find the sub-tasks corresponding to edges that: (1) lie before $p$ in a path from $q_0$ to any $q \in F$, and, (2) do not lie in a path to $q \in F$ that does not contain $p$. Intuitively, if we already have a set of policies that can generate a successful trajectory to reach the node $p$, we do not need to learn policies for sub-tasks that ultimately lead *only* to $p$ (e.g., in Fig. 1 if we have successful policies for Task$(q_0, q_2)$ and Task$(q_2, q_3)$, we can discard Task$(q_0, q_1)$ and Task$(q_1, q_3)$). We add all such sub-tasks to the set of Discarded Tasks DT (line 17), and set the Teacher Q-values for all the discarded tasks to $-\infty$ to prevent them from being sampled for the Student agent (line 18). As an extension, in the limit, an optimal policy can be found by not completely discarding such sub-tasks, but rather biasing away from them so that they would still be explored.

**Traversing in the DAG until $\phi$ satisfied:** Subsequently, we determine the next set of tasks Tasks$(\overline{E}_{AT})$ in the DAG to add to the AT set (line 19). This is calculated by identifying sub-tasks corresponding to all the outgoing edges from $p$.

Since the edge $e_{q,p}$ corresponds to the transition $q \xrightarrow{b_{(q,p)}} p$, we have a successful policy that can produce a trajectory that reaches a state where $b_{(q,p)}$ hold true, and Tasks$(\overline{E}_{AT})$ corresponds to $\mathcal{X}[p] \backslash$DT ('$\backslash$' refers to set-minus) i.e., sub-tasks corresponding to all the outgoing edges from $p$

Algorithm 1: *LSTS* ( $\mathcal{G}_\phi, M, \eta, x$ )

**Output**: Set of learned policies : $\Pi^*$, Edge-Policy Dictionary $\mathcal{P}$

1: **Placeholder Initialization**:
2: Sets of: Active Tasks ($\text{AT}$) $\leftarrow \emptyset$;
   Learned Tasks ($\text{LT}$) $\leftarrow \emptyset$; Discarded Tasks ($\text{DT}$) $\leftarrow \emptyset$
3: Edge-Policy Dictionary $\Pi$ : $\text{Task}(e) \rightarrow \pi$
4: Teacher Q-Value Dictionary: $Q$ : $\text{Task}(e) \rightarrow -\infty$
5: **Algorithm:**
6: $\mathcal{X} \leftarrow \text{Adjacency\_Matrix} (\mathcal{G}_\phi)$
7: $\text{AT} \leftarrow \text{AT} \cup \{\text{Tasks}(\mathcal{X}[q_0])\}$
8: $\forall\, \text{Task}(e) \in \text{AT} : Q[e] = 0$
9: **while** True **do**
10: $\quad e \leftarrow \text{Sample}(Q)$
11: $\quad \Pi[e], g \leftarrow \text{Learn}(M, \mathcal{G}_\phi, e, x, \mathcal{P})$
12: $\quad \text{Update\_Teacher}(Q, e, g)$
13: $\quad$ **if** $\text{Convergence}(Q, e, g, \eta)$ **then**
14: $\quad\quad \Pi^* \leftarrow \Pi^* \cup \Pi[e]$ ; $\text{LT} \leftarrow \text{LT} \cup \{\text{Task}(e)\}$ ;
      $\text{AT} \leftarrow \text{AT} \setminus \{\text{Task}(e)\}$
15: $\quad\quad Q[e] = -\infty$
16: $\quad\quad \text{Tasks}(\overline{E}_{DT}) \leftarrow \text{Discarded\_Tasks}(\mathcal{X}, e)$
17: $\quad\quad \text{DT} \leftarrow \text{DT} \cup \text{Tasks}(\overline{E}_{DT})$
18: $\quad\quad \forall\, \text{Task}(\overline{e}) \in \text{Tasks}(\overline{E}_{DT}) : Q[\overline{e}] = -\infty$
19: $\quad\quad \overline{E}_{AT} \leftarrow \text{Next\_Tasks}(\mathcal{X}, e, \text{DT})$
20: $\quad\quad$ **if** $|\text{Tasks}(\overline{E}_{AT})| = 0$ **then**
21: $\quad\quad\quad$ break
22: $\quad\quad$ **end if**
23: $\quad\quad \forall\, \text{Task}(\overline{e}) \in \text{Tasks}(\overline{E}_{AT}) : Q[\overline{e}] = 0$
24: $\quad\quad \text{AT} \leftarrow \text{AT} \cup \text{Tasks}(\overline{E}_{AT})$
25: $\quad$ **end if**
26: **end while**
27: **return** $\Pi^*, \Pi$

that do not lie in the $\text{DT}$ set.

After identifying $\text{Tasks}(\overline{E}_{AT})$, we set Teacher Q-values for all $\text{Task}(\overline{e}) \in \text{Tasks}(\overline{E}_{AT})$ to 0 so that the Teacher will sample these tasks (line 23). In our episodic setting, the episode always starts from a state $s \sim \mathcal{S}_0$ where the propositions for $q_0$ hold true, and if the current sampled sub-task is $\text{Task}(p, r)$, the agent follows a trajectory using learned policies from $\Pi^*$ to reach a state where the propositions for reaching DAG node $p$ hold true (i.e., $s \in \beta(p)$). The agent then attempts learning a separate policy for $\text{Task}(p, r)$.

The above steps (lines 9-26) go on iteratively until $|\text{Tasks}(\overline{E}_{AT})| = \emptyset$. This indicates we have no further tasks to add to our sampling strategy, and we have reached a node $q \in F$. Thus, we break from the while loop (line 21) and return the set of learned policies $\Pi^*$, and edge-policy dictionary $\Pi$ (line 27). From $\Pi$ and $\Pi^*$, we get an ordered list of policies $\Pi^*_{list} = [\pi_{(q_1,q_2)}, \pi_{(q_2,q_3)}, \ldots, \pi_{(q_{k-1}, q_k)}]$ such that sequentially following $\pi \in \Pi^*_{list}$ generates trajectories that satisfy the SPECTRL objective $\phi$ [2].

**Guarantee**: Given the ordered list of policies $\Pi^*_{list}$, we can generate a trajectory $\zeta$ in the task $M$ with

---

[2] Link to code to be provided after review

---

$\text{Pr}_{\zeta \in \mathcal{Z}}[\zeta \text{ satisfies } \phi] \geq \eta$ (Details in Appendix B).

## 5 Experimental Results

We aim to answer the following questions: (Q1) Does *LSTS* yield sample efficient learning compared to state-of-the-art baselines? (Q2) After reaching a sub-task goal state, can we sample a new sub-task to continue training and improve sample efficiency? (Q3) Does *LSTS* yield sample efficient learning for complex robotic tasks with partially observable or continuous control settings? (Q4) How does *LSTS* scale to complex search-and-rescue scenarios?

### 5.1 LSTS - Gridworld Domain

To answer (Q1), we evaluated *LSTS* on a Minigrid (Chevalier-Boisvert, Willems, and Pal 2018) inspired domain with the SPECTRL objective:

$$\phi_f^{grid} := (\texttt{achieve}(k_1) \texttt{ or } \texttt{achieve}(k_2);$$
$$\texttt{achieve}(d); \texttt{achieve}(g))\texttt{ensuring}(\neg l) \quad (6)$$

where $k_1, k_2, d, g, l$ represent the atomic propositions $Key_1, Key_2, Door, Goal, Lava$ respectively. The environment and its $\phi$ are given in Fig. 1. Essentially, the agent needs to collect *any* of the *Keys* before heading to the *Door*. After *toggling* the *Door* open, the agent needs to visit the grid with the *Goal*. At all times, the agent needs to avoid the *Lava* object. We assume an episodic setting where an episode ends if the agent touches the *Lava* object, reaches the *Goal* or exhausts the number of allocated interactions.

This is a complex problem as the agent needs to perform a series of correct actions to satisfy $\phi_f^{grid}$. The agent has access to three navigation actions: *move forward*, *rotate left* and *rotate right*. The agent can also perfom: *pick-up* action, which adds the *Key* to the agent's inventory if it is facing the *Key*, *drop* places the *Key* in the next grid if *Key* is present in the inventory, and, *toggle* that toggles the *Door* (closed $\leftrightarrow$ open) only if the agent is holding the *Key*. In this experiment, we assume a fully-observable setting where the environmental state is a low-level image encoding of the state. For each cell in the grid, the low-level encoding returns an integer that describes the item occupying the grid, along with any additional information (e.g., the *Door* can be open or closed).

For the Student RL agent, we use PPO (Schulman et al. 2017), which works for discrete and continuous action spaces. We consider a standard actor-critic architecture with 2 convolutional layers followed by 2 fully connected layers. For *LSTS*, the reward function is sparse. The agent gets a reward of $(1 - 0.9 \frac{(\text{interactions taken})}{(\text{interactions allocated})})$ if it achieves the goal in the sub-task, and a reward of 0 otherwise. For sub-tasks, $interactions\ allocated = 100$. The agent does not receive any negative rewards for hitting the $Lava$.

**Baselines:** We compare our *LSTS* method with six baseline approaches: learning from scratch (LFS), Reward Machine-based (RM) baselines: GSRS (Camacho et al. 2018), QRM (Icarte et al. 2018); and Compositional RL from Logical Specifications (DɪRL) (Jothimurugan et al. 2021). All the baselines are implemented using the RL algorithm (PPO), described above. GSRS assigns reward inversely

| Approach | # Interactions (Mean ± SD) | Success Rate (Mean ± SD) |
|---|---|---|
| $LSTS^{ct}$ | $(5.75\pm0.38)\times10^6$ | $0.96 \pm 0.02$ |
| $LSTS$ | $(6.12\pm0.25)\times10^6$ | $0.95 \pm 0.01$ |
| $\text{DIRL}^c$ | $(7.97\pm0.46)\times10^6$ | $0.95 \pm 0.03$ |
| $\text{DIRL}$ | $(9.62\pm0.42)\times10^6$ | $0.94 \pm 0.01$ |
| QRM | $5 \times 10^7$ | $0.05 \pm 0.04$ |
| GSRS | $5 \times 10^7$ | $0 \pm 0$ |
| TSCL | $5 \times 10^7$ | $0 \pm 0$ |
| LFS | $5 \times 10^7$ | $0 \pm 0$ |

Table 1: Table comparing #interactions & success rate. *LSTS* (highlighted) outperfomed all baselines

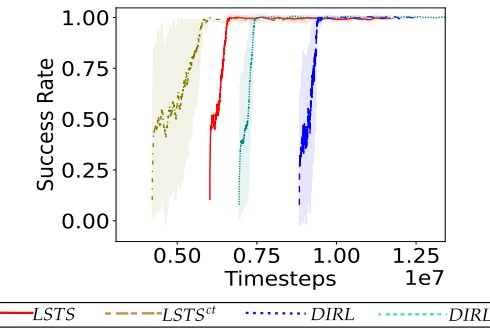

Figure 2: Averaged over 10 trials: Learning curves for approaches whose policies successfully converged.

proportional to the distance from the RM goal node. QRM employs a separate Q-function for each node in the RM, and DIRL uses Dijkstra's algorithm (edge cost is the average RL policy success rate) to guide the agent in choosing a path from the specification graph. For the fifth baseline, we modify DIRL such that instead of manually specifying a limit on the number of interactions, which needs to be fine-tuned to suit the task, we stop learning a sub-task once it has reached the convergence criteria defined in Section 4. We call this modified baseline as $\text{DIRL}^c$. The sixth baseline (TSCL (Matiisen et al. 2020)) follows a curriculum learning strategy where the Teacher samples most promising task without the use of any automaton to guide the learning progress of the agent. (More details in Appendix C)

The results in Table 2 and Fig. 2 (averaged over 10 trials) show that *LSTS* reaches a successful policy quicker compared to all baselines. $LSTS^{ct}$ is a modified version of *LSTS* and is described in Sec. 5.1. The learning curves in Fig. 2 have an offset on the x-axis to account for the interactions in the initial sub-tasks before moving on to the final task in the specification, signifying *strong transfer* (Taylor and Stone 2009). Our custom baseline, $\text{DIRL}^c$ is more sample-efficient than DIRL, and both outperform other baselines, which do not learn a meaningful policy. We performed an unpaired t-test (Kim 2015) to compare *LSTS* against the best performing baselines at the end of $10^7$ training steps and we observed statistically significant results (95% confidence). Thus, *LSTS* not only achieves a better success rate, but also converges faster (statistical significance result details in Appendix D). Time-to-threshold metric is defined as the difference in number of interactions between two approaches to reach a desired performance (Narvekar et al. 2020). From Fig. 2, we see that the time-to-threshold between *LSTS* and the best-performing baseline $\text{DIRL}^c$ is $1.85 \times 10^6$ interactions for 95% success rate.

**LSTS$^{ct}$ (LSTS + Cont. Training) - Gridworld Domain**
In *LSTS*, while learning a policy for $\text{Task}(q, p)$, we reinitialize the environment to a random initial environmental state $s \sim \mathcal{S}_0$ once the agent reaches a state where the propositions ($b_{(q,p)}$) hold true. To answer the question Q2, instead of

resetting the environment after reaching such a state where $b_{(q,p)}$ hold true, we let the Teacher agent sample a task (let's say $\text{Task}(p,r)$) from the set $\mathcal{X}[p] \setminus \text{DT}$, where $\mathcal{X}$ is the adjacency matrix for the graph, and DT is the set of Discarded Tasks, as defined in Algo. 1. This helps the agent continue its training by attempting to learn a policy $\pi_{(p,r)}$ for $\text{Task}(p,r)$ while simultaneously learning a separate policy $\pi_{(q,p)}$ for $\text{Task}(q,p)$. If the agent fails to satisfy $\text{Task}(p,r)$, we reset the environment to state $s \sim \mathcal{S}_0$. Otherwise, the agent continues its training until its trajectory satisfies the high-level objective $\phi$. We call this approach $LSTS^{ct}$ (Detailed algo in Appendix A). Results in Table 2 and Fig. 2 demonstrate that this approach improves sample efficiency by reducing the number of interactions required to learn a successful policy for the gridworld task, with a time-to-threshold metric of $3.7 \times 10^5$ interactions as compared to *LSTS*.

### 5.2 LSTS and LSTS$^{ct}$ - Robotic Domains

To answer (Q3), we test *LSTS* and $LSTS^{ct}$ on two simulated robotic environments with high interaction cost. The task in Fig. 3a has the following SPECTRL objective:

$$\phi_f^{navigation} := (\texttt{achieve}(Key_1) \, \texttt{or} \, \texttt{achieve}(Key_2);$$
$$\texttt{achieve}(Goal)) \, \texttt{ensuring}(\neg Lava) \quad (7)$$

In this task, the agent (a simulated TurtleBot) needs to collect any of the keys (yellow blocks) present in a $[3m, 3m]$ continuous environment before reaching the goal position (gray block). At all times, the agent needs to avoid the lava object (red wall) present in the center. The move forward (backward) action causes the robot to move forward (backward) by $0.1m$ and the robot rotates by $\pi/8$ radians with each rotate action. The pick-up and drop actions have effects similar to the gridworld domain. The robotic domain is more complex as objects can be placed at continuous locations. The agent receives an ego-centric image view of the environment (top-right corner of Fig. 3a), which makes the task partially observable in nature and more complex to get a successful policy. The RL agent is described in Sec. 5.1.

The second environment (Fig. 3c) consists of a simulated robotic arm pushing two objects to their target locations (Gallouédec et al. 2021) with the SPECTRL formula:

$$\phi_f^{manipulation} := \texttt{achieve}(p_1) \, ; \, \texttt{achieve}(p_2) \quad (8)$$

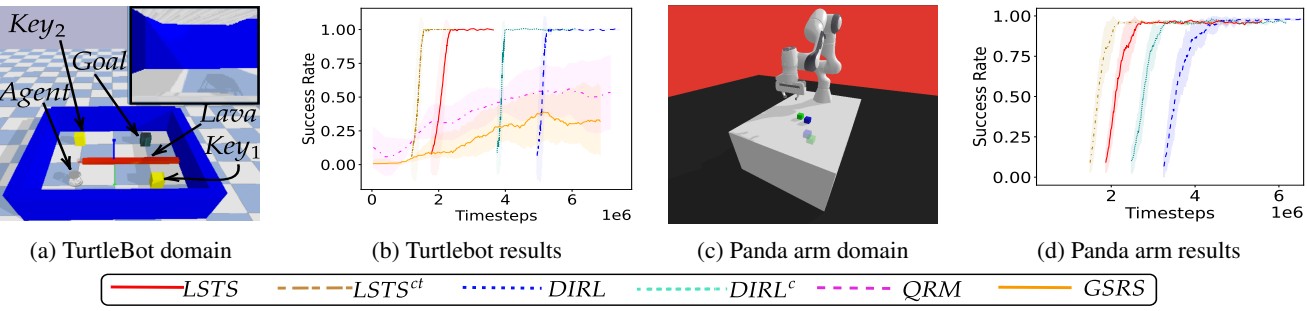

| (a) TurtleBot domain | (b) Turtlebot results | (c) Panda arm domain | (d) Panda arm results |

LSTS ——  LSTS$^{ct}$ – – – –  DIRL ·······  DIRL$^c$ ·······  QRM – – –  GSRS ——

Figure 3: Learning curves (Averaged over 10 trials) for the two robotic domains.

where $p_1$ and $p_2$ are the atomic propositions for *'push-object-1'*, *'push-object-2'*. The robot has continuous action parameters for moving the arm and a binary gripper action (close/open). An episode begins with the two objects randomly initialized on the table, and the robotic arm has to push these two objects to its final location. The agent receives its current end-effector pose, positions and velocities of the two objects, and the desired goal position for the two objects. For this task, we use the Deep Deterministic Policy Gradient with Hindsight Experience Replay (DDPG-HER) (Andrychowicz et al. 2017) as our RL algorithm. DDPG-HER is implemented using the OpenAI baselines (Dhariwal et al. 2017). Both the robotic domains were modeled using PyBullet (Coumans and Bai 2021), and the reward structure for both the RL agents was sparse, similar to the one described in Sec. 5.1. The learning curves for the TurtleBot domain (Fig. 3b) and the Panda arm domain (Fig 3d) (averaged over 10 trials) are shown in Fig. 3b and Fig. 3d respectively. For both domains, *LSTS* outperforms all the baselines in terms of learning speed. *LSTS$^{ct}$* further speeds-up learning for both the robotic domains. The time-to-threshold between *LSTS* and the best performing baseline (our custom implementation) DIRL$^c$, is $2 \times 10^6$ for the TurtleBot domain and $5 \times 10^5$ for the Panda arm domain.

### 5.3 LSTS - Search and Rescue task

To demonstrate how LSTS performs when the plan length becomes deeper, we evaluated LSTS on a complex urban Search and Rescue domain with multi-goal objectives. In this domain, the agent acts in a grid setting where it needs to perform a series of sequential sub-tasks to accomplish the final goal of the task. The agent needs to open a door using a key, then collect a fire extinguisher to extinguish the fire, and then find and rescue stranded survivors. The order in which these individual sub-goals such as opening the door, rescuing the survivors, and extinguishing the fire are achieved does not matter. A fully-connected graph $\mathcal{G}_{phi}$ generated using the above mentioned high-level states consists of 24 distinct DAG paths. This is a multi-goal task as the agent needs to find the key to open the door, then extinguish fire and rescue survivors to reach the goal state (details in Appendix F). The results in Table 2 (averaged over 10 trials) show that *LSTS* reaches a successful policy quicker compared to the LFS, GSRS, QRM and TSCL. The overall number of in-

| Approach | # Interactions (Mean $\pm$ SD) | Success Rate (Mean $\pm$ SD) |
|---|---|---|
| *LSTS* | $(8.61\pm0.12)\times10^6$ | $0.87 \pm 0.04$ |
| *LFS* | $5 \times 10^7$ | $0 \pm 0$ |
| GSRS | $5 \times 10^7$ | $0.05 \pm 0.04$ |
| QRM | $5 \times 10^7$ | $0.05 \pm 0.04$ |
| TSCL | $5 \times 10^7$ | $0 \pm 0$ |

Table 2: Table comparing #interactions & success rate for the Search and Rescue domain.

teractions to learn a set of successful policies for satisfying the high-level goal objective are higher compared to the door key experiment because of the additional complexity of task. We observe that LSTS is able to accommodate the task and learn RL policies that satisfy the high-level goal objective.

## 6  Conclusion

We proposed *LSTS*, a framework for dynamic task sampling for RL agents using the high-level SPECTRL objective coupled with the Teacher-Student learning strategy. Through experiments, we demonstrated that *LSTS* accelerates learning, converging to a desired success rate quicker as compared to other curriculum learning and automaton-guided RL baselines. *LSTS$^{ct}$* further improves sample efficiency by continuing exploration on a new sub-task once a goal state for a sub-task is reached. We also evaluate our approach on long-horizon complex robotic tasks where the state space is large and the actions are continuous. *LSTS* reduces training time without relying on human-guided dense reward function, accelerating learning when the high-level objective is available.

**Limitations & Future Work:** In certain cases, the SPECTRL objective can be novel and/or generating the labeling function can be infeasible. Our future plans involve expanding our framework to scenarios where obtaining a precise SPECTRL specification is challenging. As an extension, we would like to explore biasing away from sub-tasks rather than completely discarding them once the target node is reached, so in the limit, optimal policies can be obtained. We would also like to explore complex robotic and multi-agent scenarios with elaborate SPECTRL objectives.

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
