# OpenReview forum: "Logical Specifications-guided Dynamic Task Sampling for Reinforcement Learning Agents"
_icaps-conference.org/ICAPS/2024/Conference — ICAPS 2024_

### Official Review · Reviewer_VXmJ · 2024-01-21

**Significance And Importance:** 2
**Soundness:** 3
**Novelty:** 2
**Clarity:** 4
**Confidence:** 5

**Weaknesses:**

-1: Major weaknesses requiring significant work to be addressed for the paper to be accepted.

**Contributions Of The Paper:**

This paper proposes LSTS, a specification guided task sampling method by incorporating episodic MDP. By converting the agent’s goal to a Spectrl specification, the framework generates a DAG to formulate sub-goals associated with each higher-level goal. Then, a teacher is responsible for always selecting the most optimal sub-goals and discarding less promising sub-goals. Based on the samples provided by the teacher, the student interacts with the environment and learns to perform the task. The main contribution of this paper is formalizing the workflow of both the student and the teacher based on the DAG structure of Spectrl specification formulas. Additionally, the paper proposes criteria for identifying and discarding unpromising sub-tasks. The proposed method is evaluated against scenarios with three different Spectrl specifications.

**Ethical Considerations:**

(1) Not Applicable: The paper does not have any ethical considerations to address

**Nomination For Best Paper:**

No

**Overall Evaluation:**

-1: (weak reject)

**Questions For Authors:**

•	The paper needs more discussion on closely related existing works. For example, Shukla, Y., Gao, W., Sarathy, V., Velasquez, A., Wright, R., & Sinapov, J. (2023). LgTS: Dynamic Task Sampling using LLM-generated sub-goals for Reinforcement Learning Agents. arXiv preprint arXiv:2310.09454.
•	In the experiment scenario where the order of sub-tasks does not matter (section 5.3 and table 2), how does the performance of the proposed method compare to DIRL?
•	Can the proposed method be extended to multi-agent scenarios? Specifically, how can the Spectrl language be utilized to decompose and specify corporative goals among agents?
•	In line 585, did you mean “Table 1 and Fig. 2”?

**Reproducibility:**

3: Authors describe the implementation and domains in sufficient detail.

**Strengths Of The Paper:**

+ The proposed method converts the entire goal to a Spectrl formula, then creates a corresponding DAG. Therefore, the method can avoid unreachable states and infeasible scenarios even before sampling sub-tasks, such that computational resources are effectively conserved.
+ The writing is clear and easy to follow. The paper is well-motivated.
+ The proposed framework achieves better performance compared to other methods, such as DIRL.

**Weaknesses Of The Paper:**

- The novelty of this paper is incremental. The paper needs more discussion on closely related existing works.
- The ability of the proposed method depends partly on the expressive capability of the Spectrl specification language. Specifically, the Spectrl language can only be used to express disjunction, always, eventually, and sequentially (phi 1 then phi 2). This limits the versatility of the proposed method for more advanced features.
- The Spectrl specification language itself is not timed. Therefore, it is unable to specify goals such as “the robot should reach goal 1 in 10 seconds”. There exist more advanced specification languages such as STL and MTL that can express timing constraints but are not considered in the paper.
- There are three evaluation cases in total, while two of them are highly similar (sections 5.1 and 5.2). It is not clear what other more complex scenarios can be achieved using the Spectrl language and the proposed training framework.

---

### Official Review · Reviewer_ZNgb · 2024-01-23

**Significance And Importance:** 3
**Soundness:** 3
**Novelty:** 3
**Clarity:** 4
**Overall Evaluation:** 2
**Confidence:** 3

**Weaknesses:**

2: No major or minor weaknesses.

**Contributions Of The Paper:**

The paper introduces Logical Specifications-guided Dynamic Task Sampling (LSTS), a learning approach that improves the sampling efficiency in learning reinforcement learning policies for complex sequential decision problems. It is a dynamic task sampling technique that uses high-level task specifications to guide the learning process of agents. The authors evaluate the approach through experiments in different settings, demonstrating its effectiveness against benchmark approaches.

**Ethical Considerations:**

(1) Not Applicable: The paper does not have any ethical considerations to address

**Nomination For Best Paper:**

No

**Questions For Authors:**

1) A prerequisite for using LSTS is to first formalize the learning problem with SPECTRL formulas. What if my planning problem is so complex that it is difficult for humans to formalize? It would be beneficial if the authors could discuss any potential limitations of their approach. Specifically, are there scenarios where LSTS is not applicable due to the complexity of formalizing the problem? Such a discussion would provide a clearer understanding of the method's applicability and limitations.

2) In section 5.1, line 585, the text references "Table 2 and Fig. 2". It seems this might be an error, and it should refer to "Table 1"?

3) In the supplementary material, the source code is labeled as "Automaton-Guided Dynamic Task Sampling for Reinforcement Learning Agents (AGTS)” instead of LSTS. Could the authors clarify the reason for this ? Is the source code for a different method or an earlier version of LSTS? Is AGTS a published method?

Remark upon rebuttal: I would like to thank the authors, I appreciate that you include the discussion in the camera ready version. My rating remains the same.

**Reproducibility:**

4: Authors promise to release code and domains (whichever apply).

**Strengths Of The Paper:**

- The paper provides clear and comprehensive explanations of LSTS, making it accessible and informative for readers.
- The paper includes an extensive experimental evaluation comparing LSTS with baseline approaches in three different scenarios, demonstrating the effectiveness and transferability of the method.
- The paper addresses the well-known problem of sample efficiency in reinforcement learning, and presents a solution that improves learning efficiency in complex environments.
- The provision of source code in the supplementary material is a major strength, facilitating reproducibility of the work.

**Weaknesses Of The Paper:**

- Minor weakness: lack of discussion of limitations of the approach (see comment 1)

---

> ### Author Rebuttal · Authors · 2024-01-28
>
> We thank the reviewer for their thoughtful comments.
>
> 1) (a) SPECTRL allows users to encode tasks through sequences, disjunctions and conjunctions of subtasks and also specify safety properties. This covers a wide range of problems commonly encountered by the RL community. Certain problems might require tighter bounds, and even more expressivity in the human-specified formulas. Linear Temporal Logic (LTL), Signal Temporal Logic (STL)  and Branching-Time Logics are few approaches that are more expressive than SPECTRL and can specify complex problems. However, the DFA representation of LTL can contain cycles making it difficult to be used for RL problems where sample efficiency is crucial. This is because few sub-tasks in the DFA representation would correspond to transitions in the DFA that reach a node farther from the accepting node of the DFA, and this is not desirable as it would lead to more interactions in the environment. Other temporal logic approaches are even more complex to express, and this is why such approaches are not commonly used for specification-guided RL problems. Further studies need to be directed toward using timed and more expressive languages for specification-guided RL problems by the community.
> We will add a discussion on these approaches by the camera-ready version of the paper.
>
> 1. (b) The other aspect of this question, concerning tasks that are too complex to be formalized, is considerably harder to solve using specification-guided RL approaches. RL with sparse reward settings might be able to achieve the goal in such situations, but that would lead to inefficient solutions in terms of environmental interactions. Another way is to attempt to learn the specification through successful trajectories in the environment or through expert demonstrations would be  interesting to study but tangential to the scope of the proposed work.
>
> 2) Thank you for pointing out the typo. We will change it to Table 1.
>
> 3) Thank you for pointing out the typo. We apologize for the name of the method being AGTS in the readme of the code. AGTS is not a published method. We initially wanted to use complex specifications that could be expressed using automatons expressed using LTL, but later updated the specification language to be more coherent. The readme and the code will be updated before being open sourced.

---

### Official Review · Reviewer_iGdH · 2024-01-28

**Significance And Importance:** 2
**Soundness:** 4
**Novelty:** 3
**Clarity:** 4
**Overall Evaluation:** 2
**Confidence:** 3

**Weaknesses:**

2: No major or minor weaknesses.

**Contributions Of The Paper:**

This paper clearly explains the idea (LSTS), it appears to be novel and sensible. LSTS builds on a previous approach, called DIRL, that uses this logical task specification language to obtain a higher-level policy using planning/search in an abstract MDP composed of subgoals and RL to learn the low-level policies to achieve these subgoals. The key idea in this work is to avoid learning policies for many possible subtasks at the lower-level, by both focusing on the subtasks where the policy is learning most effectively (highest return) and a discarding subtasks to avoid wasting samples on them. This idea is a useful extension on DIRL and empirically provides the purported benefits (improved sample efficiency over DIRL). I recommend acceptance. But, I have a few minor comments below to help improve the paper.

**Ethical Considerations:**

(1) Not Applicable: The paper does not have any ethical considerations to address

**Nomination For Best Paper:**

No

**Questions For Authors:**

Remark: None of the following are questions that the authors are required to respond to. They are largely for constructive criticism. I would appreciate a response to some of the more concrete points below, but definitely the authors do not need to respond to musings about alternative choices for learning progress.

The Teacher-Student structure is quite interesting. Really, this is simply a bi-level structure, where the Teacher adjusts which subtask to learn based on Student progress, so that jointly they find a set of successful policies for the agent to reach a goal state. The terminology, in my opinion, somewhat conflicts with the way Student-Teacher is typically used, for example, in the distillation literature. This is relatively minor, and I am not saying this must be changed, just pointing it out.

It took quite some time to learn what the rewards were for the lower-level agent. Eventually, I say this one sentence: “The Student agent receives a positive reward for satisfying b(q0 ,q1 ) and a small negative reward at all other time steps. ” So the subtasks are all cost-to-goal problems. This should be stated earlier. Or, you could also consider asking if this is necessary. Why not allow the rewards to be more general?

Nonetheless, this important detail came late and led me to be confused about certain choice. For example, I had the following question, which was clarified later once I saw that there are cost-to-goal subtasks.
“epsilon-greedy for task selection is not particularly focused on learning progress. What if one task is actually quite difficult to accomplish, but has high rewards initially? Then the gt would be high, but the agent has not made that much progress.”
I see how this is not an issue now with cost-to-goal problems. But thus it is more important to state this focus on cost-to-goal subtasks more clearly upfront so the reader is not left filling in guessed details.

It would also be nice to talk more about alternative measures of learning progress. Returns are a reasonable choice. But, did you consider others? Again, not required here, since the primary goal for this paper I believe is to introduce this formalism, where other choices for Q[e] can be considered in follow-up work.

This formalism is new to me and I had a few motivational questions. What settings can we expect the designer to be able to specify subtasks effectively in this language? Can the agent be sure it is always feasible to achieve the subtasks? What if the designer provided a specification where it is not possible?

“We add all such sub-tasks to the set of 480 Discarded Tasks DT (line 17), and set the Teacher Q-values
for all the discarded tasks to −∞ to prevent them from being sampled for the Student agent (line 18).”
This does not seem right, since epsilon-greedy would still select this task. Presumably, you do not need to set the value to negative infinity, since it is simply not considered when the teacher takes actions.

**Reproducibility:**

5: Code and domains (whichever apply) are already publicly available

**Strengths Of The Paper:**

- Novel idea
- Comprehensive experiments

**Weaknesses Of The Paper:**

- Could have explained the rewards for the subtasks earlier
- Could more clearly outline how much this builds on DIRL. The introduction does a good job explaining the key limitation of DIRL, and that you address it, but it could be good to highlight if there are other key deviations beyond carefully selecting which subtasks to learn on.

---

### Meta-Review · Area_Chair_64jz · 2024-02-06

**Recommendation:** Accept (Poster)
**Confidence:** 5

**Metareview:**

The paper is overall of a good quality, fairly novel, well written and relevant to ICAPS. Some issues were raised about literature positioning and the expressivity of SPECTRL, plus some additional ones which can be regarded as minor. The major ones appear to have been addressed satisfactorily in the author response, and it is reasonable to expect that, in case of acceptance, the authors will incorporate their response in the final version of the paper, as well as fixing the minor issues. Overall, the work meets the standards of an ICAPS paper.

**Ethical Considerations:**

(1) Not Applicable: The paper does not have any ethical considerations to address